# Umbrella Review of Systematic Reviews and Meta-Analyses on the Consumption of Different Food Groups and the Risk of Overweight and Obesity

**DOI:** 10.3390/nu17040662

**Published:** 2025-02-13

**Authors:** Emilie Kristoffersen, Sofie Lassen Hjort, Lise M. Thomassen, Elaheh Javadi Arjmand, Matteo Perillo, Rajiv Balakrishna, Anindita Tasnim Onni, Ida Sofie Karlsen Sletten, Antonello Lorenzini, Lars T. Fadnes

**Affiliations:** 1Department of Global Public Health and Primary Care, University of Bergen, 5009 Bergen, Norway; e.kristoffersen@student.uib.no (E.K.); sofie.hjort@student.uib.no (S.L.H.); elaheh.javadi@uib.no (E.J.A.); rajiv.balakrishna@student.uib.no (R.B.); anindita.onni@uib.no (A.T.O.); lars.fadnes@uib.no (L.T.F.); 2Department of Addiction Medicine, Haukeland University Hospital, 5009 Bergen, Norway; 3Department of Biomedical and Neuromotor Sciences, University of Bologna, 40126 Bologna, Italy; matteo.perillo2@unibo.it (M.P.); antonello.lorenzini@unibo.it (A.L.); 4Medical Faculty, University of Bergen, 5009 Bergen, Norway; ida.sletten@uib.no; 5Biostructures and Biosystems National Institute (INBB), 00136 Rome, Italy

**Keywords:** food, nutrition, diet, food groups, dietary pattern, overweight, obesity, umbrella review

## Abstract

Background/Objectives: Dietary choices play an essential role in energy balance and weight gain. This systematic umbrella review investigates the association between the intake of various food groups (whole grains, refined grains, fruits, vegetables, nuts, legumes, fish, eggs, total dairy, red meat, white meat, processed meat, added sugar, and sugar-sweetened beverages) and their associations to the risk of obesity and being overweight. Methods: We systematically searched Medline, Embase, Web of Science, and Epistemonikos for systematic reviews and meta-analyses. A total of 2925 articles were screened, and 13 articles were included in our analysis. Results: For each food group, data included a mean of 166,100 participants and 36,760 cases, ranging from 19,885 participants and 7183 cases for red meat to 520,331 participants and 91,256 cases for nuts. Heterogeneity was high for most of the food groups except for whole grains and sugar-sweetened beverages. The quality of the most comprehensive meta-analyses was high for all food groups, except for processed meats, which was of low quality. High intakes of whole grains, legumes, nuts, and fruits are associated with a reduced risk of overweight and obesity. In contrast, high intakes of red meat and sugar-sweetened beverages are associated with increased risk of overweight and obesity. No significant results were found for the remaining food groups, and no meta-analysis was found for fish, eggs, white meat, and added sugars. Conclusions: Diets rich in whole grains, legumes, nuts, and fruits are associated with a lower risk of developing obesity and being overweight. In contrast, diets high in red meat and sugar-sweetened beverages are associated with an increased risk of overweight and obesity.

## 1. Introduction

In recent decades, nutritional research has moved from the single-nutrient approach to a more complete understanding of people’s diets, often categorized by food groups. A healthy diet encompasses a combination of foods that people should generally increase their intake of, as well as foods that should be limited, to promote health and reduce the risk of diet-related chronic diseases [1,2,3].

What we eat and drink represents a person’s total energy intake. An imbalance between energy intake and energy expenditure, resulting in an energy surplus over time, causes weight gain, overweight, and obesity. Obesity is associated with a lower quality of life and a reduced lifespan [4]. In addition, obesity is identified as the most important risk factor associated with developing several chronic diseases, including type 2 diabetes mellitus and cardiovascular disease [5,6], as well as a major risk factor for the development of several cancers [7]. Current obesity management strategies have generally failed to prevent the obesity epidemic from escalating, causing more than 2.8 million deaths each year [8,9].

National dietary guidelines provide information to the general population on healthy eating. Many countries provide public guidance on healthy eating through quantitative advice on how much or how little one should include in the diet of different food groups. This is generally based on research examining the relationship between diet and chronic diseases [10]. A high-quality meta-analysis was conducted five years ago [11], but substantial research has been conducted since then. Thus, there is no up-to-date systematic overview of how intakes of all food groups are associated with overweight and obesity. Understanding the relationship between food groups and overweight/obesity is key for informing future public health strategies and must be based on the highest quality of evidence. To our knowledge, there is no systematic umbrella review providing up-to-date evidence on the association between food groups and overweight/obesity. Umbrella reviews summarize systematic reviews and meta-analyses, providing condensed information on a research-dense topic and the highest quality of evidence currently available within a research topic [12]. The aim of the current study was to perform a systematic umbrella review to update the evidence on the association between fourteen predefined food groups (whole grains, refined grains, fruits, vegetables, nuts, legumes, fish, eggs, total dairy, red meat, white meat, processed meat, added sugar, and sugar-sweetened beverages) and their associations to the risk of overweight and obesity using the umbrella review framework.

## 2. Materials and Methods

The umbrella review framework was used to summarize all available evidence from meta-analyses and systematic reviews examining the associations between the intake of different food groups and the incidence of overweight and obesity. The study protocol was registered at PROSPERO (CRD42024557902).

Eligibility criteria: Meta-analyses and systematic reviews presenting analyses from longitudinal observational studies (e.g., cohort and case-control) and trials on adults (age ≥ 18 years) investigating the association between food groups as the exposure and the incidence of overweight and obesity as the outcome were considered eligible. The food groups included were whole grains, refined grains, fruits, vegetables, nuts, legumes, fish, eggs, total dairy, red meat, white meat, processed meat, added sugar, and sugar-sweetened beverages. Details on the serving sizes are presented in Appendix A. Comparators were high versus low consumption and per-serving estimates for the dose-response relationship between the exposures and the outcome. Studies were excluded if they included a cross-sectional design (unless providing estimates for cohorts and/or case-control studies when excluding cross-sectional studies), or providing regional estimates not generalizable to a broader population (e.g., studies only conducted in one single country). Papers were also excluded if they only reported on macro- or micronutrients, were animal studies, or were non-systematic reviews. Articles in languages other than English were also excluded.

Search: A systematic literature search was performed using predefined terms in Medline, Embase, Web of Science, and Epistemonikos from database inception to June 2024. Detailed search strategy is presented in Appendix A. After automatic deduplication using EndNote 21, all remaining references were imported into AS Review for screening.

Screening: The screening process involved reviewing titles and abstracts, followed by a full-text assessment of relevant articles. The title and abstract screening were conducted using AS Review v.1.6.3r0 [13]. AS Review is an active-learning-powered tool that learns from assessments performed by the reviewer and iteratively provides an updated list of records more likely to be relevant. The screening process was performed by two authors (EK, SLH) independently. Disagreements were resolved by consensus with a third author (EJA). The screening was then double-checked by two authors (EJA, LMT).

Data extraction: The data extraction process involved gathering information on the following variables: first author, title, exposure (food group), outcome (overweight and obesity incidence), population (age), study design, search year, search month, number of studies, number of participants, number of cases/outcomes, outcome measures, heterogeneity, findings (high versus low intake or per serving), and range of the non-linear dose-response curves. Data were extracted independently by two authors (EK, SLH) and double-checked by two other authors (EJA, LMT).

Quality assessment: The quality assessment of each publication was performed using the validated A Measurement Tool to Assess Systematic Reviews (AMSTAR) 2, which categorizes articles as high, moderate, low, or critically low quality [13]. Two authors (EK, SLH) rated the studies, and their ratings were checked by two other authors (EJA, LTF).

Most relevant meta-analysis: For each combination of food groups and type of comparison (high versus low and per serving), the most relevant meta-analysis was identified by selecting the most recent and comprehensive meta-analysis with the highest number of studies, participants, or cases of overweight/obesity. Identification of the most relevant meta-analysis was done for each combination of food groups and type of comparison (high versus low and per serving) to identify the most recent and comprehensive meta-analysis. This was done by selecting the most recent among the meta-analyses that had the highest number of studies, participants, or cases of overweight/obesity. The studies graded as low or critically low quality were excluded by the pool of candidates unless no high-quality meta-analyses were available for that specific combination of food group and comparison.

Data analysis: Data handling, generation of tables, and plots were performed using R 4.4.1. The results of the reviewed meta-analyses were summarized in forest plots that visualize the association between the consumption of food groups and the incidence of overweight and obesity. Additionally, tables were generated with the following information: first author, search year, number of primary studies, number of participants, number of observed cases, estimated association with overweight/obesity (typically as relative risk ratios with 95% confidence intervals), heterogeneity (for high versus low and per serving), and AMSTAR-2 score. The results presented as odds ratios (OR) were converted to relative risk ratios (RR). The conversion equations for OR to RR can be found in Appendix A. The serving sizes used in the dose-response analysis can be found in Appendix A. For high versus low and per-serving comparison, the same data were used to produce forest plots that summarize the associations reported in the meta-analyses. Results and characteristics from the most recent and comprehensive meta-analyses are presented in this paper as forest plots and tables. Tables presenting the results of all meta-analyses can be found in Appendix A.

## 3. Results

### 3.1. Search and Screening

A total of 2925 articles were identified and screened by title and abstract, resulting in 31 potentially eligible articles that were included in the full-text reading. During the phase of full-text readings, the reasons for exclusions were as follows: not a systematic review (n = 1), not a relevant outcome (n = 13), not a relevant exposure (n = 1), conference abstract (n = 1), book (n = 1). Data extraction was performed for 14 articles. During the data extraction process, one additional article was excluded due to the cross-sectional design. This left us with 13 articles included in this umbrella review [11,14,15,16,17,18,19,20,21,22,23,24,25]. This included eight systematic reviews with meta-analyses and five systematic reviews without meta-analysis. The details of the search and identification of studies are presented in the PRISMA flow chart (Figure 1) and the search string can be found in Appendix A.

### 3.2. Systematic Reviews with Meta-Analysis

#### 3.2.1. Overweight/Obesity

Seven systematic reviews with meta-analyses investigated the association between food groups and the incidence of overweight/obesity combined. The median number of primary studies included per meta-analysis was four (ranging from 1–14). All the articles included cohort studies, and one article included a randomized controlled trial. None of the reviewed meta-analyses included case-control primary studies. The studies were conducted between 2013 and 2021. The mean number of participants and cases for all food groups combined was 166,100 and 36,760, respectively, ranging from 19,885 participants and 7183 cases for red meat to 520,331 participants and 91,256 cases for nuts. Heterogeneity for most studies was higher than 0.5, meaning that the observed association between these food groups and the incidence of overweight/obesity is quite diverse across the primary studies included in the meta-analyses. For whole grains and sugar-sweetened-beverages (both high versus low and per-serving), heterogeneity was less than 0.5, indicating a less diverse association. Legumes (both high versus low and per-serving) and red meat (high versus low) are based on only one study. The meta-analyses on high versus low comparisons and per- serving showed that high intakes of whole grains, nuts, legumes, and fruits were associated with a lower risk of overweight/obesity. A non-significant trend was found for vegetables. In contrast, higher intakes of red meat and sugar-sweetened beverages were associated with a higher risk of overweight/obesity, and non-significantly for processed meats and refined grains. The dose-response relationships for these food groups revealed distinct patterns. Refined grains and sugar-sweetened beverages were associated with gradually increased risk with higher intakes. Vegetables followed a U-shaped curve, with moderate consumption associated with the lowest risk, but not as high for a very high intake. Moderate intake of nuts was associated with reduced risk of obesity, which attenuated with higher intakes. Whole grains and fruits were associated with reducing risk with increasing intakes across the reported intake range. For dairy, the dose-response curve was ambiguous, with an increase from small to moderate intake associated with an increased risk of obesity/overweight, while a high intake was inversely associated. No significant associations were found between intakes of total dairy and overweight/obesity, either in the high versus low comparison or per-serving relationships (Figure 2 and Figure 3). Results from all included articles are presented in Appendix A.

#### 3.2.2. Obesity

Five systematic reviews with meta-analyses investigated the association between food groups and the incidence of obesity. All included primary studies were cohorts. The median number of primary studies per meta-analysis was 4.5 (ranging from 2–5), and the studies were conducted between 2013 and 2021. The mean number of participants for all food groups combined was 103,161, ranging from 12,987 participants for sugar-sweetened beverages to 254,779 participants for nuts [19]. Heterogeneity was higher than 0.5 for nuts and processed meat and 0.36 for sugar-sweetened beverages. For nuts, heterogeneity was particularly high (0.89), indicating that the association between nut consumption and the risk of obesity vary much between the included studies. The intake of sugar-sweetened beverages was significantly associated with a higher risk of obesity in both the high versus low comparison and linear dose-response. In contrast, nut consumption was associated with lower risk of obesity in the linear dose-response. High intake of processed meat is potentially associated with an increased risk of obesity, but the results are non-significant. Figures visualizing these results can be found in Appendix A.

### 3.3. Systematic Reviews Without Meta-Analyses

In addition to the identified meta-analyses presented above, five systematic reviews without meta-analyses presented data on the association between the intake of different food groups and the incidence of overweight and obesity [21,22,23,24]. A systematic review by Eslami et al. of six studies shows that long-term nut consumption may be associated with less weight gain and reduced risk of being overweight/obese [24]. Four out of six studies showed an inverse association between nut consumption (typically at dosages of one to two servings per week) and weight gain, as well as the risk of overweight/obesity. The remaining two studies focused on the association between nut consumption and changes in waist circumference, but only one of these found a significant inverse association.

Nine studies in the systematic review by Louie and colleagues focused on adults, and five of these showed a negative association between dairy consumption and weight gain [21]. One study found a negative association to obesity only among men who were initially overweight. Two studies showed both a reduced and increased risk, depending on the type of dairy product, while one study found no association between dairy consumption and weight gain. The article concludes that existing evidence is insufficient to establish a clear link and points to challenges related to methodological variations in the included studies as a main reason for the conflicting findings.

In the systematic review by Vasanti and colleagues, four studies focused on adults, and one of them found a significant association between the consumption of sugar-sweetened beverages and weight gain in women [22]. Another study reported a non-significant association in both genders. However, this study included a smaller sample size, which may have reduced its ability to detect significant effects.

A systematic review by Nour et al. examined the association between vegetable intake and weight-related outcomes in adults [23]. Ten studies were included in the analysis, and six of these showed that increased intake of vegetables was associated with weight loss or a reduced risk of weight gain and/or overweight/obesity. Four studies only measured vegetable intake at baseline and found varying associations, including an inverse association with waist circumference in women and a reduced risk of weight gain. The article concluded that there is moderate evidence for an inverse association between vegetable intake and weight-related outcomes in adults.

A systematic review by Trumbo and Rivers concluded that the association between sugar-sweetened beverages and obesity is inconsistent when adjusting for energy balance [25].

### 3.4. Quality Assessment

Among the 13 articles included, reporting on the risk of overweight/obesity and obesity [11,14,15,16,17,18,19,20,21,22,23,24,25], six were rated as high-quality using AMSTAR-2, two articles were rated as moderate, one as low, and five as critically low. The quality criteria that mostly contributed to downgrading were not reporting funding of included studies, not having registered protocols, and not justifying exclusions. For the most comprehensive and recent studies, all food groups were available from high-quality articles, except for processed meat, which was graded as low. The quality assessment scoring for all articles is presented in Appendix A.

## 4. Discussion

### 4.1. Main Findings

To our knowledge, this is the first umbrella review summarizing available evidence from meta-analyses and systematic reviews of prospective observational studies on the association between food groups and the risk of overweight and obesity, using the umbrella review framework. We summarize estimates for associations with overweight and obesity for comparisons of high versus low, per serving, and non-linear dose-response relationships of whole grains, refined grains, nuts, legumes, fruits, vegetables, sugar-sweetened beverages, red meat, processed meat, and total dairy. It is well known that maintaining a healthy body weight is important to lower the risk of obesity-related comorbidities such as diabetes, cardiovascular disease, and many types of cancers [26]. Our umbrella review provides evidence that high intakes of whole grains, legumes, nuts, and fruits are associated with a reduced risk of overweight and obesity, while red meat and sugar-sweetened beverages are associated with an increased risk of overweight and obesity.

### 4.2. Comparisons with Dietary Guidelines and Public Health Implications

Our findings align with nutritional guidelines emphasizing the importance of including plenty of fruits, legumes, nuts, and whole grains as part of a healthy diet. These foods are abundant in essential nutrients important for maintaining body functions [27], and are associated with reduced mortality, disability-adjusted life years lost, and increased life expectancy [2,3,10]. Additionally, the association between high intakes of red meats and sugar-sweetened beverages and increased risk of overweight and obesity found in this umbrella review was not surprising, as lowering intakes of unhealthy food items high in energy and saturated fats is often included in public health strategies promoting healthy eating to the general population [28]. Many dietary guidelines recommend limiting snacks such as cakes, as snacks and some highly processed foods are associated with obesity [29,30]. We have used a food group approach, as most foods can be decomposed into food groups. For example, consuming cake as a snack typically can be broken into sugar, milk, refined grains, and eggs. Many of the foods associated with a reduced risk of obesity are plants with low carbon footprints, which aligns with the finding that sustainable food choices are associated with a reduced risk of obesity [31].

High heterogeneity was observed for several food groups. Possible reasons for this include differences in study design and methods, variation in study participants, differences in adjustment for confounding variables, and different interventions or exposures. Another possible reason for the high heterogeneity could be that some of the studies are carried out in different geographical regions, leading to variations in dietary habits and other lifestyle factors. For instance, regarding processed meat, some studies identified differences between men and women, as well as variations in the amount of processed meat consumed. Also, some studies adjusted for energy intake (which might be unfortunate in this case, as this is likely to mediate the effect), while other studies did not. Some studies included participants with overweight in the assessment of obesity. Additionally, some studies had a predominance of healthcare professionals who may not be representative of the general population. There were also variations in the type of nuts the participants were exposed to. In some studies, one single type of nut was used, while others used mixed nuts. The nut dose also ranged from 5–100 g/day. High heterogeneity makes conclusions less certain and limits generalizability.

### 4.3. Biological Plausibility

Fruits, legumes, nuts, and whole grains are all rich in dietary fibers, which could lower body weight by stimulating satiety [32]. A possible mechanism includes gastric distention following the high hydration properties of viscous fibers, which activates vagal afferents to the brain, stimulating the secretion of satiety hormones that facilitate a reduction in eating frequency and overall food consumption [33]. A systematic review of associations between the intake of dietary fibers and body weight indicated improvements in body weight associated with dietary fiber consumption, in addition to improvements in risk factors for diet-related chronic diseases, such as insulin sensitivity and cholesterol levels [33,34]. One previous umbrella review has investigated the association between the Mediterranean diet and weight gain [35]. They found that adhering to Mediterranean diets was associated with a lower risk of developing obesity in adults. Similar findings have also been observed for children [36].

The association between the consumption of red meat and increased risk of developing obesity could be explained by the high fat content found in red meat, which contributes to high energy density associated with weight gain. In line with this, some research indicates that obesity is linked to lower taste sensitivity and a higher preference for and intake of fatty foods and, to a lesser extent, sweet foods [32]. In addition, the structure of the fatty acids may be of importance, as red meat has a high content of saturated fatty acids, which are considered obesogenic and have been linked to the development of diet-related chronic diseases such as cardiovascular disease [37,38]. Sugar-sweetened beverages also provide abundant energy but few essential nutrients. Clinical trial data support our findings and provide strong evidence that sugar-sweetened beverages cause weight gain, possibly by providing high caloric content to the diet and inducing hyperinsulinemia through rapid glucose absorption [39]. Studies on the impact of different types of sugars (including fructose, sucrose, and glucose) on metabolic outcomes have shown no substantial differences between the type of sugars and weight gain [40,41]. This suggests that the association we have found could be related to a surplus of energy rather than the fructose syrup that is often used in sugar-sweetened beverages.

The dose-response relationship between several of the food groups and the risk of overweight and obesity are assessed through non-linear associations. Refined grains are associated with an increased risk of obesity/overweight with increasing intake levels. This could possibly be explained by the fact that refined grains have a rapid digestion, a high glycemic index, and overconsumption of foods high in refined grains could lead to an energy surplus and fat storage [42,43]. Associations for vegetables could suggest that optimal consumption levels for many could be in the range around 250–400 g per day [11]. The non-linear associations for nut consumption and overweight/obesity indicated a reduced risk of obesity for intakes around 10 g per day, attenuates with higher intake levels. On the other hand, evidence from trials indicates higher nut consumption does not appear to cause greater weight gain and may even have a protective effect on weight control, even with higher intakes. Nevertheless, their calorie density may offset their benefits when overconsumed [11]. Our findings from the non-linear dose-response curves illustrate the nuanced relationships between food intake and obesity risk, underlining the importance of balanced consumption and highlight the need for tailored dietary guidelines to effectively address obesity and its related health concerns.

### 4.4. Strengths and Limitations

This is the most comprehensive umbrella review to date investigating the relationship between food groups and overweight/obesity, with no other umbrella review of similar scope currently available. The umbrella design summarizes all related systematic reviews and meta-analyses, providing the most up-to-date and comprehensive evidence on associations between food group intake and the risk of overweight or obesity. To ensure high-quality research, we followed a pre-registered PROSPERO protocol, adhered to PRISMA guidelines, and performed screening, extractions, and quality assessments in duplicates. Limitations of this umbrella review include the fact that for some of the food groups, such as legumes and red meat, few studies were available, contributing to uncertainty. Furthermore, heterogeneity was generally high for most food groups except whole grains and sugar-sweetened beverages, which might indicate that there is substantial variation within the food groups related to associations with the risk of developing overweight and obesity.

### 4.5. Future Research

Although many studies included in the current umbrella review adjusted for age, alcohol consumption, and physical activity, this assessment does not summarize non-dietary factors and their associations with overweight and obesity. In addition, no applicable data was available on the association between the intake of fish, eggs, and white meat and their associations to overweight and obesity, which would be important for future research. Alcoholic beverages could also be obesogenic and contribute to energy intake but are not considered in our approach, nor are other substances or drugs. Nevertheless, the food groups we include generally constitute most of the energy intake for most people.

## 5. Conclusions

The findings from this umbrella review indicate that diets rich in whole grains, legumes, nuts, and fruits are associated with a lower risk of developing overweight or obesity. A similar trend was also observed for vegetables. On the other hand, higher consumption of red meat and sugar-sweetened beverages is associated with an increased risk of developing overweight and obesity, with a non-significant trend for refined grains and processed meats. These findings generally align with dietary guidelines and strengthen the evidence for recommending diets rich in plant-based foods while limiting the intake of sugar-sweetened beverages and red meat. Since there is substantial heterogeneity for most food groups, contributing to some uncertainty, the findings should be interpreted with some caution. Further studies are needed to investigate the risk of obesity with different foods within each food group. For whole grains and sugar-sweetened beverages, heterogeneity is lower, and the quality of the background evidence is high. No systematic reviews provided data on the association between the intake of fish, eggs, white meat, and added sugars with the risk of overweight and obesity. Further studies that categorize food groups while focusing on weight measures could shed light on the risk of obesity and consumption patterns.

## Figures and Tables

**Figure 1 nutrients-17-00662-f001:**
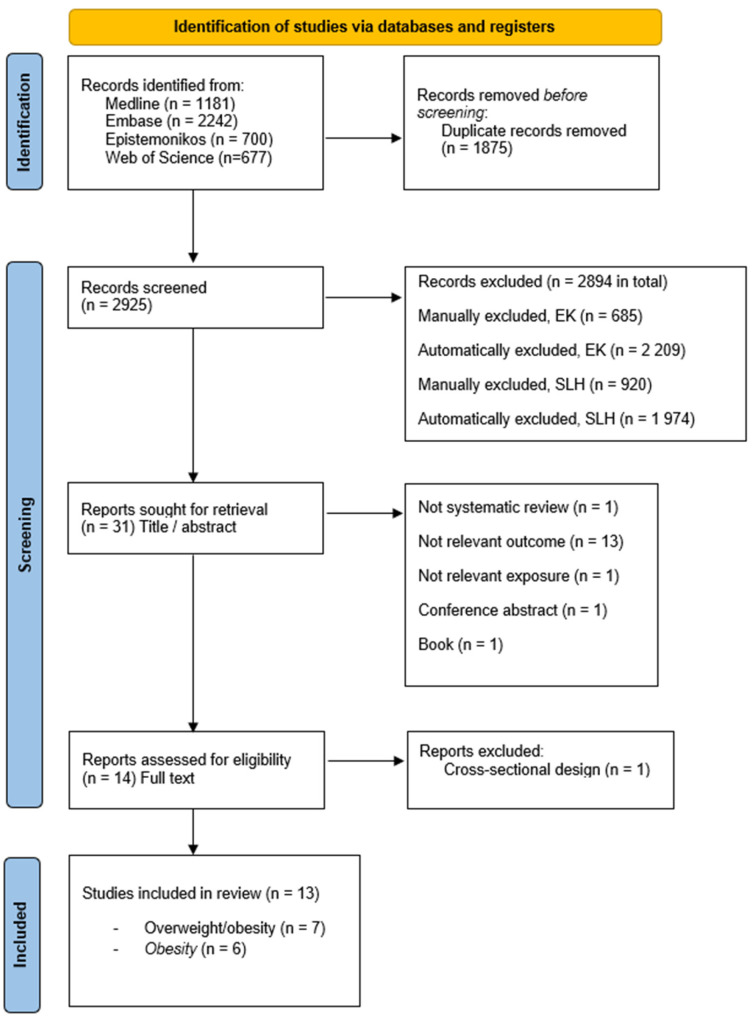
PRISMA flow chart describing the process of identifying, screening, and including eligible studies.

**Figure 2 nutrients-17-00662-f002:**
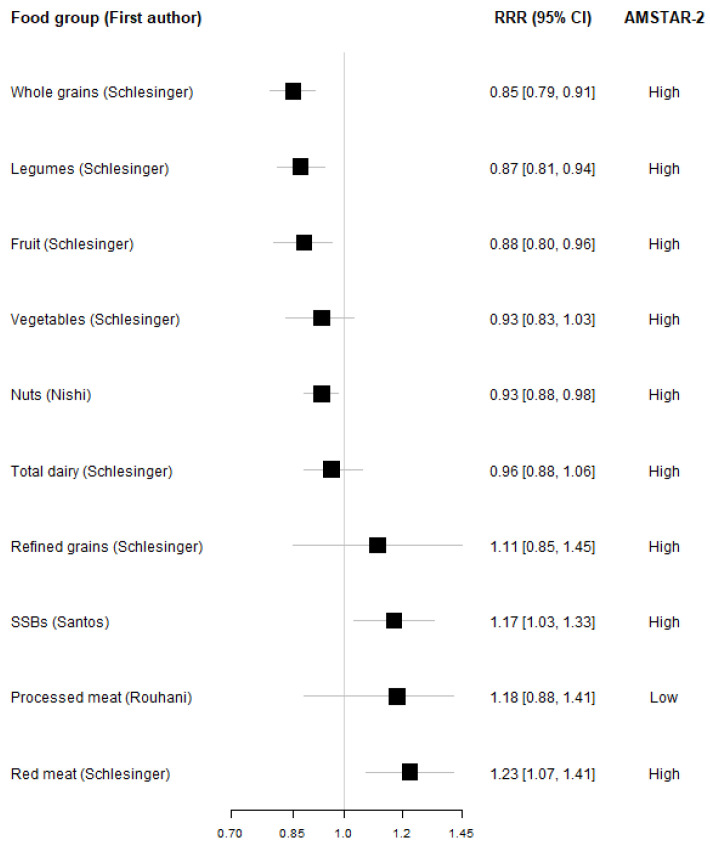
Associations between food groups (high versus low) and the incidence of overweight and obesity in the most comprehensive and up-to-date meta-analyses.

**Figure 3 nutrients-17-00662-f003:**
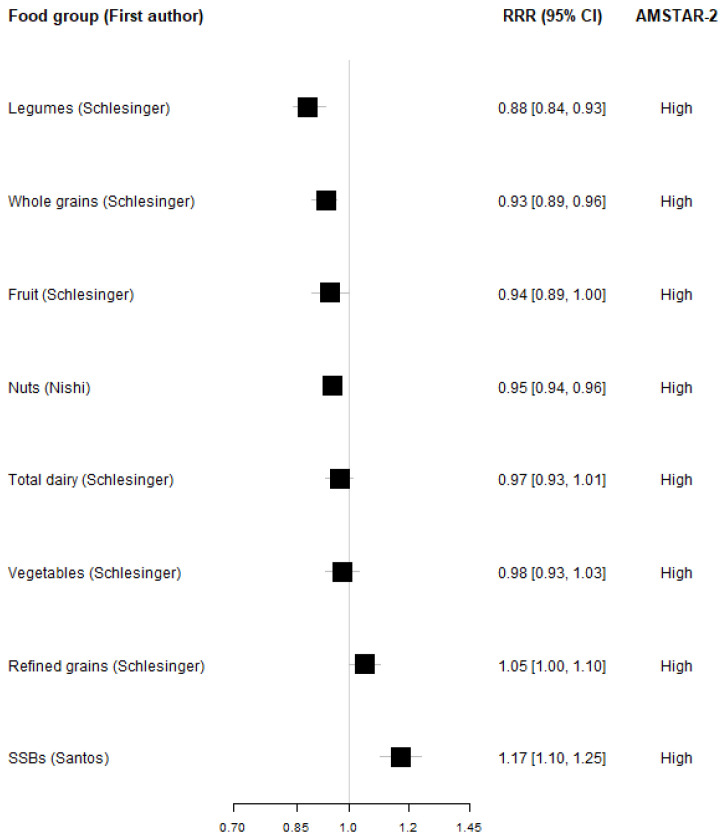
Linear dose-response associations between food groups (per serving) and the incidence of overweight and obesity in the most comprehensive and up-to-date meta-analyses. The following serving sizes were used: whole grain: 30 g; vegetables 100 g; fruits 80 g; legumes 50 g; nuts 28 g; refined grains 30 g; sugar-sweetened beverages 250 mL; total dairy 200 g.

## Data Availability

Data is available in Appendix A.

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
