# Peer review of "Umbrella Review of Systematic Reviews and Meta-Analyses on the Consumption of Different Food Groups and the Risk of Overweight and Obesity"

_nutrients, 2025, doi:10.3390/nu17040662_

Round 1
Reviewer 1 Report
Comments and Suggestions for Authors
Please check attached file

Author Response
Comment 1: Lacks a clear aim of the work.
Response 1: Thank you for pointing this out. We agree that the aim was not sufficiently clear and have now tried to be clearer and more precise in our section on the aim of the work. The changes are marked red and can be found on page 2, lines 65-70. It now reads: “Umbrella reviews summarize systematic reviews and meta-analyses, providing condensed information within a research dense topic, and provide the highest quality of evidence currently available within a research topic [10]. The aim of the current study was to perform a systematic umbrella search of the literature to update the evidence on the association between fourteen pre-defined food groups (whole grains, refined grains, fruits, vegetables, nuts, legumes, fish, eggs, total dairy, red meat, white meat, processed meat, added sugar and sugar-sweetened beverages) and their associations to risk of overweight and obesity using the umbrella review framework.”
Comment 2: Have snacks and alcoholic beverages been evaluated? If not, why not? These are major problems often resulting from overconsumption.
Response 2: Thank you for your comment. We have used the food groups approach in our search string as most of the foods we eat can be decomposed into food groups providing most of the daily energy intake. We agree that snacks are relevant for overconsumption and in our assessment that is inherently included, usually as a combination of refined grains, added sugar, and dairy. For example, consuming cake as a snack typically can be broken into sugar, milk, refined grains, and eggs. Regarding alcoholic beverages, these are generally not considered in our approach as are not other substances or drugs, since they require a different categorization than the food groups we included. For instance, some alcoholic beverages like beer and wine are often consumed with meals in larger portions, while beverages with higher alcohol content like spirits are consumed in smaller portions in different settings. This makes it difficult to compare to other included food groups as they have been reported as grams per day consumption. We have added some discussion about these aspects.
Comment 3: The authors reported that they included 13 articles with obese or overweight subjects. One would think that one could provide in a table the information that was included in the cited articles, giving BMI, weight, and in addition physical activity, age, region of life.
Response 3: Thank you for pointing this out. We agree that providing information about body mass index, weight, age, physical activity and region of life from the included articles could be useful. We have added additional columns for age and country in table A2-A4 in the appendix, page 6-11. However, only Nishi 2019 reported body weight and BMI, while none reported on physical activity level. Therefore, we did not add additional columns for weight, BMI and physical activity.
Comment 4: That is, what kind of research? I guess the authors mean follow-up a the same category. That is just my guess, which probably other readers would also like to understand.
Response 4: Thank you for this comment, we are not sure what sentence you are referring to since the comment has appeared for the whole page 9 of the manuscript. If there is confusion about what type of research we have done, this is an umbrella review or a review of the reviews that have been done on this theme. We have provided some more details in the introduction related to this. If the question refers to what type of research is needed to address the heterogeneity of some results, often cohort studies with subcategorization of food groups can answer the following questions.
Reviewer 2 Report
Comments and Suggestions for Authors
This umbrella review aims to synthesize evidence on the association between various food group consumption and the risk of overweight and obesity. While the topic is relevant, and the systematic approach is commendable, several areas require attention to enhance the rigor and clarity of the manuscript.
1. The manuscript should carefully distinguish between association and causation throughout. Phrases suggesting causality should be replaced with more appropriate terms like "associated with" or "linked to," reflecting the observational nature of the included studies.
2. The high heterogeneity observed for several food groups warrants a more in-depth discussion. Explore potential sources of heterogeneity and discuss its implications for the certainty of the findings.
3. The conclusion should be nuanced regarding food groups not included in meta-analyses (e.g., fish, eggs, white meat, added sugars). Avoid definitive statements of "no significant results" and instead emphasize the lack of sufficient evidence.
4. The discussion of complex dose-response relationships (e.g., U-shaped curves) needs further elaboration on potential biological mechanisms and clinical relevance.
5. Restructure the Discussion section with clear subheadings to address key aspects such as main findings, comparison with existing guidelines, biological plausibility, strengths and limitations, future research, and public health implications.
6. Strengthen the Conclusion by incorporating more specific, actionable recommendations based on the findings, while acknowledging the limitations.
7. Explicitly state the novelty and value of this umbrella review in the Discussion section.
Author Response
Comment 1: The manuscript should carefully distinguish between association and causation throughout. Phrases suggesting causality should be replaced with more appropriate terms like "associated with" or "linked to," reflecting the observational nature of the included studies.
Response 1: Thank you for addressing this. We agree with your comment and have made changes to the results section and discussion section to make it clearer that we are looking at associations and have avoided language implying causal relationships. The changes are marked with red and can be found at pages 8-10, line 217, 224, 225, 232, 262.
Comment 2: The high heterogeneity observed for several food groups warrants a more in-depth discussion. Explore potential sources of heterogeneity and discuss its implications for the certainty of the findings.
Response 2: Thank you for pointing this out. We agree with this comment. Therefore, we have included more discussion related to the high heterogeneity at page 9, lines 290-305. It now reads: “High heterogeneity was observed for several food groups. Possible reasons for this include differences in study design and methods, variation in study participants, differences in adjustment for confounding variables, and different interventions or exposures. Another possible reason for the high heterogeneity could be that some of the studies are carried out in different geographical regions, which may lead to variations in dietary habits and other lifestyle factors. For instance, regarding processed meat, there were identified differences between men and women in some of the studies, and there were also variations in the amount of processed meat consumed in the different studies. Also, some studies adjusted for energy intake (which might be unfortunate in this case as this is likely to mediate the effect), while other studies did not. Some studies included participants with overweight in the assessment of obesity. Additionally, some studies had a predominance of healthcare professionals who may not be representative of the general population. There were also variations in the type of nuts the participants were exposed to. In some studies, one single type of nut was used, while mixed nuts were used in others. Also, the nut dose ranged from 5-100 g/day. High heterogeneity makes conclusions less certain and limits generalizability.”
Comment 3: The conclusion should be nuanced regarding food groups not included in meta-analyses (e.g., fish, eggs, white meat, added sugars). Avoid definitive statements of "no significant results" and instead emphasize the lack of sufficient evidence.
Response 3: Thank you for pointing this out. We agree with your comment. We have now added to the discussion a conclusion for the food groups not involved in the meta-analysis. This is marked red and can be found on pages 9-10, lines 359-361. In the conclusion, we added “No systematic reviews provided data on the association between the intake of fish, eggs, white meat, and added sugars with the risk of obesity and being overweight. Further studies with categorization of food groups focusing on weight measures can shed light on the risk of obesity and consumption patterns.”
Comment 4: The discussion of complex dose-response relationships (e.g., U-shaped curves) needs further elaboration on potential biological mechanisms and clinical relevance.
Response 4: We agree with your comment that this section was confusing and insufficiently linked to the mechanisms. We have revised this section and tried to link this better to the biological mechanisms. The findings are marked in red and can be found on page 10, lines 336-353. It is now read: “Refined grains are associated with an increased risk of obesity/overweight with increasing intake levels. This could possibly be explained by the fact that refined grains have a rapid digestion and high glycemic index, and overconsumption of foods high in refined grains could lead to energy surplus and fat storage [43, 44]. Associations for vegetables could suggest that optimal consumption levels for many could be in the range around 250-400 grams per day [21]. The non-linear associations for nut consumption and overweight/obesity indicated a reduction of risk of obesity for intakes around 10 grams per day attenuating with higher intake levels. On the other hand, evidence from trials indicates higher nut consumption does not appear to cause greater weight gain, and possibly even having a protective effect in weight control even with higher intakes. Nevertheless, their calorie density may offset their benefits when overconsumed [21]. Our findings from the non-linear dose response curves illustrate the nuanced relationships between intake of food and obesity risk, underlining the importance of balanced consumption and highlight the need for tailored dietary guidelines to effectively address obesity and its related health concerns.”
Comment 5: Restructure the Discussion section with clear subheadings to address key aspects such as main findings, comparison with existing guidelines, biological plausibility, strengths and limitations, future research, and public health implications.
Response 5: Thank you for pointing this out. We have now restructured the discussion section added subheadings as you have suggested and added additional information under these headings where this was needed. The changes are marked in red in the revised manuscript on pages 8-11, line numbers 258-378.
Comment 6: Strengthen the Conclusion by incorporating more specific, actionable recommendations based on the findings, while acknowledging the limitations.
Response 6: Thank you for your comment. We have now added more to the conclusion about limitations and future research which can be found in lines 392-396.
Comment 7: Explicitly state the novelty and value of this umbrella review in the Discussion section.
Response 7: Thank you for your comment. We have now added more about the value of the study in discussion: Line 361: “The umbrella design summarizes all the systematic reviews and meta-analyses addressing related research questions, which provides the most up-to-date and comprehensive evidence on associations between intake of food groups and risk of obesity or being overweight.”
Reviewer 3 Report
Comments and Suggestions for Authors
Dear Authors,
This study was conducted to umbrella review of systematic reviews and meta-analyses on consumption of different food groups and risk of overweight and obesity. I am sorry that as you can see, the subject of this study is a very simple one, and there are already many similar studies that have been systematically and well reported around the world (Please refer to below articles).
1. Spinelli S, Monteleone E. Food Preferences and Obesity. Endocrinol Metab (Seoul). 2021 Apr;36(2):209-219. doi: 10.3803/EnM.2021.105. Epub 2021 Apr 19. PMID: 33866777; PMCID: PMC8090462.
2. Garipağaoğlu M, Sahip Y, Budak N, Akdikmen O, Altan T, Baban M. Food types in the diet and the nutrient intake of obese and non-obese children. J Clin Res Pediatr Endocrinol. 2008;1(1):21-9. doi: 10.4008/jcrpe.v1i1.5. Epub 2008 Aug 4. PMID: 21318061; PMCID: PMC3005637.
3. Schlesinger, S., Neuenschwander, M., Schwedhelm, C., Hoffmann, G., Bechthold, A., Boeing, H., & Schwingshackl, L. (2019). Food Groups and Risk of Overweight, Obesity, and Weight Gain: A Systematic Review and Dose-Response Meta-Analysis of Prospective Studies. Advances in nutrition (Bethesda, Md.), 10(2), 205–218.
4. Hatta M, Horikawa C, Takeda Y, Ikeda I, Yoshizawa Morikawa S, Kato N, Kato M, Yokoyama H, Kurihara Y, Maegawa H, Fujihara K, Sone H. Association between Obesity and Intake of Different Food Groups among Japanese with Type 2 Diabetes Mellitus-Japan Diabetes Clinical Data Management Study (JDDM68). Nutrients. 2022 Jul 24;14(15):3034.
5. Christoph Reger, Michael F. Leitzmann, Sabine Rohrmann, Tilman Kühn, Anja M. Sedlmeier, Carmen Jochem. Sustainable diets and risk of overweight and obesity: A systematic review and meta-analysis. First published: 11 February 2024 https://doi.org/10.1111/obr.13707
6. GBD (2017) Diet Collaborators (2019) Health effects of dietary risks in 195 countries, 1990–2017: a systematic analysis for the Global Burden of Disease Study 2017. Lancet 393(10184):1958–1972.
7. Matthews, V.L., Wien, M. & Sabaté, J. The risk of child and adolescent overweight is related to types of food consumed. Nutr J 10, 71 (2011). https://doi.org/10.1186/1475-2891-10-71
8. Raquel de Deus Mendonça, Adriano Marçal Pimenta, Alfredo Gea, Carmen de la Fuente-Arrillaga, Miguel Angel Martinez-Gonzalez, Aline Cristine Souza Lopes, Maira Bes-Rastrollo,
9. Raquel de Deus Mendonça, Adriano Marçal Pimenta, Alfredo Gea, Carmen de la Fuente-Arrillaga, Miguel Angel Martinez-Gonzalez, Aline Cristine Souza Lopes, Maira Bes-Rastrollo. Ultraprocessed food consumption and risk of overweight and obesity: the University of Navarra Follow-Up (SUN) cohort study, The American Journal of Clinical Nutrition, Volume 104, Issue 5, 2016, Pages 1433-1440.
10. Dicken, S.J., Batterham, R.L. Ultra-processed Food and Obesity: What Is the Evidence?. Curr Nutr Rep 13, 23–38 (2024). https://doi.org/10.1007/s13668-024-00517-z
Author Response
Comment 1 This study was conducted to umbrella review of systematic reviews and meta-analyses on consumption of different food groups and risk of overweight and obesity. I am sorry that as you can see, the subject of this study is a very simple one, and there are already many similar studies that have been systematically and well reported around the world (Please refer to below articles).
1. Spinelli S, Monteleone E. Food Preferences and Obesity. Endocrinol Metab (Seoul). 2021 Apr;36(2):209-219. doi: 10.3803/EnM.2021.105. Epub 2021 Apr 19. PMID: 33866777; PMCID: PMC8090462.
2. Garipağaoğlu M, Sahip Y, Budak N, Akdikmen O, Altan T, Baban M. Food types in the diet and the nutrient intake of obese and non-obese children. J Clin Res Pediatr Endocrinol. 2008;1(1):21-9. doi: 10.4008/jcrpe.v1i1.5. Epub 2008 Aug 4. PMID: 21318061; PMCID: PMC3005637.
3. Schlesinger, S., Neuenschwander, M., Schwedhelm, C., Hoffmann, G., Bechthold, A., Boeing, H., & Schwingshackl, L. (2019). Food Groups and Risk of Overweight, Obesity, and Weight Gain: A Systematic Review and Dose-Response Meta-Analysis of Prospective Studies. Advances in nutrition (Bethesda, Md.), 10(2), 205–218.
4. Hatta M, Horikawa C, Takeda Y, Ikeda I, Yoshizawa Morikawa S, Kato N, Kato M, Yokoyama H, Kurihara Y, Maegawa H, Fujihara K, Sone H. Association between Obesity and Intake of Different Food Groups among Japanese with Type 2 Diabetes Mellitus-Japan Diabetes Clinical Data Management Study (JDDM68). Nutrients. 2022 Jul 24;14(15):3034.
5. Christoph Reger, Michael F. Leitzmann, Sabine Rohrmann, Tilman Kühn, Anja M. Sedlmeier, Carmen Jochem. Sustainable diets and risk of overweight and obesity: A systematic review and meta-analysis. First published: 11 February 2024 https://doi.org/10.1111/obr.13707
6. GBD (2017) Diet Collaborators (2019) Health effects of dietary risks in 195 countries, 1990–2017: a systematic analysis for the Global Burden of Disease Study 2017. Lancet 393(10184):1958–1972.
7. Matthews, V.L., Wien, M. & Sabaté, J. The risk of child and adolescent overweight is related to types of food consumed. Nutr J 10, 71 (2011). https://doi.org/10.1186/1475-2891-10-71
8. Raquel de Deus Mendonça, Adriano Marçal Pimenta, Alfredo Gea, Carmen de la Fuente-Arrillaga, Miguel Angel Martinez-Gonzalez, Aline Cristine Souza Lopes, Maira Bes-Rastrollo,
9. Raquel de Deus Mendonça, Adriano Marçal Pimenta, Alfredo Gea, Carmen de la Fuente-Arrillaga, Miguel Angel Martinez-Gonzalez, Aline Cristine Souza Lopes, Maira Bes-Rastrollo. Ultraprocessed food consumption and risk of overweight and obesity: the University of Navarra Follow-Up (SUN) cohort study, The American Journal of Clinical Nutrition, Volume 104, Issue 5, 2016, Pages 1433-1440.
10. Dicken, S.J., Batterham, R.L. Ultra-processed Food and Obesity: What Is the Evidence?. Curr Nutr Rep 13, 23–38 (2024). https://doi.org/10.1007/s13668-024-00517-z
Response 1: Thank you for your comment. We agree that there is substantial research that has been carried out on the consumption of food and the risk of overweight and obesity. Except for the five-year-old systematic review and meta-analysis by Schlesinger and colleagues that we also summarized, the other studies do not summarize the evidence for adults except the study focusing on sustainable eating. We are also not familiar with other studies that provide this. Thus, our motivation for carrying out an umbrella review on the topic was to summarize the substantial research available as you point out, making it easier for the readers to be up to date by providing condensed information within a research-dense topic. This is important to have available when revising dietary recommendations and policies. We agree our rationale was not sufficiently clear in the last version and with your feedback into consideration, we have added some more background for the rationale. We also added some more background for the utility of the umbrella review approach in the introduction, marked with red, page 2, lines 63-65. In addition, thank you for adding suggestions for articles. As mentioned, Schlesinger et al. was within our scope and was already included. Several of the others are relevant for discussion and we have added discussion relating to these.
Round 2
Reviewer 3 Report
Comments and Suggestions for Authors
Accept